# Evaluation of Antioxidant and Anti-α-glucosidase Activities of Various Solvent Extracts and Major Bioactive Components from the Seeds of *Myristica fragrans*

**DOI:** 10.3390/molecules25215198

**Published:** 2020-11-08

**Authors:** Cai-Wei Li, Yi-Cheng Chu, Chun-Yi Huang, Shu-Ling Fu, Jih-Jung Chen

**Affiliations:** 1Institute of Traditional Medicine, National Yang-Ming University, Taipei 11221, Taiwan; leecw1219@gmail.com (C.-W.L.); xbox88888@gmail.com (Y.-C.C.); slfu@ym.edu.tw (S.-L.F.); 2Faculty of Pharmacy, School of Pharmaceutical Sciences, National Yang-Ming University, Taipei 11221, Taiwan; jimmyhuang9289@gmail.com; 3Department of Medical Research, China Medical University Hospital, China Medical University, Taichung 40447, Taiwan

**Keywords:** *Myristica fragrans*, various solvent extracts, antioxidant activity, anti-α-glucosidase activity

## Abstract

*Myristica fragrans* is a well-known species for flavoring many food products and for formulation of perfume and medicated balm. It is also used to treat indigestion, stomach ulcers, liver disorders, and, as emmenagogue, diaphoretic, diuretic, nervine, and aphrodisiac. We examined antioxidant properties and bioactive compounds in various solvent extracts from the seeds of *M. fragrans*. Methanol, ethanol, and acetone extracts exhibited relatively strong antioxidant activities by 2,2-diphenyl-1-(2,4,6-trinitrophenyl)hydrazyl (DPPH), 2,2′-azino-bis(3-ethylbenzothiazoline-6-sulfonic acid) (ABTS), superoxide radical, and hydroxyl radical scavenging tests. Furthermore, methanol extracts also displayed significant anti-α-glucosidase activity. Examined and compared to the various solvent extracts for their chemical compositions using HPLC analysis, we isolated the ten higher content compounds and analyzed antioxidant and anti-α-glucosidase activities. Among the isolates, dehydrodiisoeugenol, malabaricone B and malabaricone C were main antioxidant components in seeds of *M. fragrans*. Malabaricone C exhibited stronger antioxidant capacities than others based on lower half inhibitory concentration (IC_50_) values in DPPH and ABTS radical scavenging assays, and it also showed significant inhibition of α-glucosidase. These results shown that methanol was found to be the most efficient solvent for extracting the active components from the seeds of *M. fragrans*, and this material is a potential good source of natural antioxidant and α-glucosidase inhibitor.

## 1. Introduction

It is widely known that active free radicals are produced by normal metabolism which cause oxidative damage to biomacromolecules, which include carbohydrates, membrane lipids, proteins, and DNA [1]. To lower the oxidative damage of active free radicals, many unnatural antioxidants, such as butylated hydroxytoluene (BHT) and butylated hydroxyanisole (BHA), with strong antioxidant activity are proverbially applied in food industry. However, recurrent discovery of potential detrimental effects, such as carcinogenesis and liver damage, by using synthetic antioxidants [2]. These evidences have raised public interest in natural antioxidants as another option. In addition, epidemiological studies have also indicated positively associated between the uptake of fruits and vegetables rich in antioxidants (e.g., phenolic compounds) and diseases prevention, such as aging, atheroscelerosis, and also cancer [3]. Therefore, researches on natural antioxidant have gained increasing concern [4,5,6]. Therefore, antioxidants from Chinese herbal medicines (CHM) have become more popular as they are low in toxicity, hardly produce complications, and have desirable pharmacological activities [7,8]. Several antioxidant attributes of Chinese herbal medicines in the context of their chemical biological mechanisms have been evaluated by Zhu et al. (2004) [9].

Disorders of carbohydrate metabolism lead to detrimental health problems, such as obesity, diabetes, and oral diseases, worldwide. Besides, vascular or metabolic disorders including hypertension and diabetes correlate to reactive oxygen species, which are crucial principal catalysts for initiating oxidative stress in vivo [10]. Improvement in cellular redox status and glycaemic control play important roles in regulating diabetes and its complications. Hyperglycemia may increase the risk of metabolic malfunctions. Along with other factors, this may result in increasing the risk of type 2 diabetes, where uncontrolled high blood glucose levels can result in several complications, such as cardiovascular complications, renal failure, blindness, and foot ulcers requiring surgical removal of limb [11,12,13].

It is crucial to manage hyperglycemia because it can induce severe complications. α-Glucosidase, which is situated on the brush-border surface membrane of intestinal cells, activates the final step of the digestive process. The absorbable monosaccharides generated by α-glucosidase activity is readily available for intestinal absorption [14]. Therefore, inhibitors of α-glucosidase restrain the release of glucose from starch, thereby reducing intestinal absorption of glucose and decreasing in postprandial hyperglycemia. Currently, there are several antidiabetic drugs, such as acarbose, miglitol, and voglibose, which act by inhibiting α-glucosidase activity to reduce high blood glucose levels. However, the continuous use of these synthetic agents often causes undesirable negative side effects, namely abdominal cramps, diarrhea, vomiting, and liver toxicity [15,16]. Past studies showed that *M. fragrans* could improve the side effects of synthetic anti-glucosidase drugs [17,18]. Therefore, there is a demand for naturally-derived α-glucosidase antidiabetic compounds without adverse or unwanted secondary effects.

*Myristica fragrans*, an important spice and medicinal plant, has been traditionally used for spice and medicinal purposes for anti-inflammatory, anxiogenic, antithrombotic, anti-ulcerogenic, antifungal, aphrodisiac, antiplatelet aggregating, antitumor, carminative, and hypolipidemic activities [19,20,21,22,23]. In our studies on the antioxidant and anti-α-glucosidase activities of plants in Taiwan, many species have been screened for these effects, and *M. fragrans* has been found to be an active species. This report depicts the evaluation of antioxidant and anti-α-glucosidase activities of various solvent extracts from the seeds of *M. fragrans* and its major bioactive components.

## 2. Results and Discussion

### 2.1. Determination of Total Phenolic Content (TPC) and Yields in Each Solvent Extract

We investigated the TPC and yields in different solvent extracts of *M. fragrans*. Table 1 shows TPC and extraction yields of *n*-hexane, chloroform, dichloromethane, ethyl acetate, acetone, methanol, and ethanol extracts from *M. fragrans*. The yields of different solvent extracts from *M. fragrans* were ranged from 15.6 ± 1.21% (ethanol extract) to 30.7 ± 1.49% (dichloromethane extract). The dichloromethane extract exhibited the highest yield among all extracts possibly due to its abundant amounts of low polar components. Significant differences were found in TPC among all different solvent extracts, of which methanol extract contained highest amount of TPC (107.83 ± 0.66 mg/g), followed by ethanol (98.01 ± 2.99 mg/g), acetone (70.07 ± 2.28 mg/g), ethyl acetate (32.93 ± 0.85 mg/g), dichloromethane (18.97 ± 1.22 mg/g), chloroform (18.65 ± 0.53 mg/g), and *n*-hexane (16.82 ± 0.62 mg/g). The higher polar methanol and ethanol extracts have the higher yield of TPC due to enriching with phenolic constituents. The low polarity solvent extracts, such as *n*-hexane, chloroform, and dichloromethane, showed less amount of TPC regardless of their high extraction yields. Extracts of methanol and ethanol showed approximately five to six times the amount of TPC in those using *n*-hexane, dichloromethane, and chloroform. These results suggested that suitable relative polarity of extracting solvents for TPC from *M. fragrans* would be ranged from 0.654 to 0.762.

### 2.2. 2,2-Diphenyl-1-(2,4,6-trinitrophenyl)hydrazyl (DPPH) Free-Radical Scavenging Activity

DPPH is relatively stable in aqueous or ethanol solution. Antioxidants could interact with DPPH radical and transfer an electron or hydrogen atom to DPPH radical so as to neutralize free radicals [24]. The DPPH radical scavenging activities of each extract were shown in Table 2, the extracts of methanol (inhibitory concentration (IC)_50_ = 22.42 ± 0.99 μg/mL) and ethanol (IC_50_ = 39.65 ± 0.83 μg/mL) exhibited great DPPH radical scavenging activities, which are comparable to that of BHT (IC_50_ = 36.94 ± 0.49 μg/mL). Besides, ethyl acetate (IC_50_ = 95.12 ± 2.63 μg/mL), dichloromethane (IC_50_ = 96.90 ± 7.68 μg/mL), *n*-hexane (IC_50_ = 126.57 ± 6.23 μg/mL), and chloroform (IC_50_ = 167.17 ± 7.13 μg/mL) extracts showed relatively high IC_50_ values.

### 2.3. 2,2′-Azino-bis(3-ethylbenzothiazoline-6-sulfonic acid) (ABTS) Free-Radical Scavenging Activity

As shown in Table 2, ethanol (IC_50_ = 27.68 ± 0.31 μg/mL) exhibited strongest ABTs radical scavenging activity, followed by methanol (IC_50_ = 34.41 ± 0.78 μg/mL), acetone (IC_50_ = 64.35 ± 1.58 μg/mL), dichloromethane (IC_50_ = 82.31 ± 2.15 μg/mL), ethyl acetate (IC_50_ = 91.19 ± 0.88 μg/mL), chloroform (IC_50_ = 93.70 ± 5.06 μg/mL), and n-hexane (IC_50_ = 103.05 ± 2.41 μg/mL).

### 2.4. Superoxide Radical Scavenging Activity

Notably, the result showed all extracts had no significant effect on superoxide radical scavenging activity (IC_50_ > 400 μg/mL) except for methanol (IC_50_ = 117.66 ± 2.56 μg/mL) (Table 2).

### 2.5. Hydroxyl Radical Scavenging Activity

This assay displays the abilities of the extracts to suppress the degradation of hydroxyl radical-mediated deoxyribose in the mixture containing FeCl_3_-ethylenediaminetetraacetic acid (EDTA)-ascorbic acid and H_2_O_2_ reaction. The extracts exhibited high scavenging activities of hydroxyl radical at low concentration and the values of IC_50_ were lower than BHT (Table 2), except of the solvents extracted by dichloromethane and chloroform. The hydroxyl radical scavenging activity of methanol (IC_50_, 37.81 ± 1.56 μg/mL) exhibited the strongest activity, followed by acetone (IC_50_, 42.99 ± 0.19 μg/mL), n-hexane (IC_50_, 51.94 ± 0.79 μg/mL), ethyl acetate (IC_50_, 55.25 ± 1.25 μg/mL), ethanol (IC_50_, 56.05 ± 2.52 μg/mL), chloroform (IC_50_, 82.39 ± 2.62 μg/mL), and dichloromethane (IC_50_, 88.19 ± 2.09 μg/mL).

Based on the above results from DPPH, ABTs, superoxide radical scavenging, and hydroxyl radical scavenging activities, methanol extract possessed the highest antioxidant activity among all solvent extracts. Among the methods used for quantifying antioxidant capacity, there were significant correlations among three assays (DPPH, ABTS, and hydroxyl radical scavenging assays), while exhibited weaker correlations with superoxide scavenging assay. Similar experimental results could also be observed in the past study [25]. This may be related to the structure and functional groups of the major active components.

### 2.6. Anti-α-glucosidase Activity Assay

Natural products which possess anti-α-glucosidase activity have received great attention because of their potential use in treating diabetes. As shown in Table 3 and Figure 1, the methanol extract of *M. fragrans* exhibited the most anti-α-glucosidase activity (IC_50_, 4.08 ± 0.12 μg/mL), followed by ethanol (IC_50_, 11.92 ± 0.39 μg/mL), acetone (IC_50_, 29.07 ± 2.30 μg/mL), ethyl acetate (IC_50_, 185.36 ± 5.21 μg/mL), *n*-hexane (IC_50_, >200 μg/mL), chloroform (IC_50_, >200 μg/mL), and dichloromethane (IC_50_, >200 μg/mL). The solvents extracted by methanol and ethanol were more effective than the positive control, quercetin (IC_50_, 14.99 ± 0.81 μg/mL). Among all solvent extracts, methanol extract showed the strongest anti-α-glucosidase activity. These results showed that higher polar solvent extracts of *M. fragrans* possessed stronger α-glucosidase inhibitory effect.

### 2.7. Quantification of Components

In previous studies, phenolic derivatives from *M. fragrans* were well identified and separated by HPLC with normal- and reversed-phase column chromatography [26,27]. The HPLC methods using both normal- and reversed-phase column for the quantification of three components were verified regarding linearity, limit of detection (LOD), and limit of quantification (LOQ). The linearity was validated by the data from six different concentrations (1.0, 5.0, 10.0, 25.10, 50.0, and 100.0 μg/mL) of the standard solutions. The linear regression parameters of calibration curves, correlation coefficient, LOD, and LOQ were shown in Appendix A. Six concentrations of each standard were analyzed in triplicate to generate respective calibration curve. The linearity (R^2^ > 0.9996) between Y (the peak area of the analytes with external standard) and X (concentration of the standards) was achieved in the tested stage. Thus, we confirmed that three major components in *M. fragrans* were malabaricone B, malabaricone C, and dehydrodiisoeugenol. Structures of three components are shown in Figure 2.

### 2.8. Quantitation of Active Components in Different Solvent Extracts

The HPLC chromatograms of each solvent extract and standard solution of *M. fragrans* in normal- and reverse-phase are displayed in Appendix A. The concentrations of three active compounds in each solvent extract is shown in Table 4. Total quantities of three active compounds in each extract was ranged from a maximum of 51.43 ± 1.18 mg/g (methanol extract) to a minimum of 15.52 ± 0.26 mg/g (*n*-hexane extract) in succeeding order of methanol > ethanol > acetone > ethyl acetate > chloroform > dichloromethane > *n*-hexane extract. Methanol (51.43 ± 1.18 mg/g), and ethanol (42.80 ± 1.17 mg/g) extracts exhibited higher amounts of phenolic compounds compared with other extracts. Furthermore, methanol extract showed approximately two times the quantity of phenolic compounds compared with acetone and ethyl acetate. Malabaricone C was the most abundant among the three phenolic compounds in organic solvent extract, followed by dehydrodiisoeugenol and malabaricone B.

### 2.9. Antioxidant Activities of Isolated Components

With the isolated malabaricone B, malabaricone C, and dehydrodiisoeugenol, we measured antioxidant activities, such as ABTS, DPPH, hydroxyl, and superoxide radical scavenging activities. Results are shown in Table 5, where malabaricone C (IC_50_, 8.35 ± 2.20 μg/mL) exhibited the strongest DPPH radical scavenging activity, followed by dehydrodiisoeugenol (IC_50_, 66.02 ± 2.85 μg/mL) and malabaricone B (IC_50_, >200 μg/mL). While all of three compounds showed high ABTS radical scavenging activities, dehydrodiisoeugenol exhibited relatively high hydroxyl radical scavenging activity. However, all of three compounds showed no significant effect on superoxide radical scavenging activity. Based on the above result, it could be inferred that the methanol extract contained the largest amount of major phenolic compounds among all extracts and thus possessed the strongest antioxidant activity.

### 2.10. Anti-α-glucosidase Activities of Isolated Component

For further evaluation of the α-glucosidase inhibitory activity, we conducted further investigation on major components isolated from *M. fragrans*. Among the isolated compounds, we found that malabaricone C possesses stronger inhibitory activity against α-glucosidase. Figure 3 shows the dose-response curves of malabaricone C on α-glucosidase inhibition with the IC_50_ value of 20.97 ± 0.17 μg/mL.

## 3. Materials and Methods

### 3.1. Chemicals and Antibodies

Folin-ciocalteau’s reagent, 2,2′-azino-bis(3-ethylbenzothiazoline-6-sulfonic acid) (abts), nicotinamide adenine dinucleotide (NADH), ascorbic acid, ethylenediaminetetraacetic acid (EDTA), α-glucosidase, and hydrogen peroxide solution were purchased from Sigma-Aldrich (St. Louis, MO, USA). Ferric chloride (FeCl_3_) and *p*-nitro-phenyl-α-*d*-glucopyranoside (*p*-NPG) were supplied by Alfa Aesar (Lancashire, UK). Potassium peroxodisulfate, sodium carbonate, potassium dihydrogenphosphate, and disodium hydrogenphosphate were purchased from the SHOWA Chemical Co. Ltd. (Chuo-ku, Japan). Loroglucinol, 2,2-diphenyl-1-(2,4,6-trinitrophenyl)hydrazyl (DPPH), phenazine methosulphate (PMS), 2-thiobarbituric acid (TBA), nitroblue tetrazolium (NBT), and deoxyribose were obtained from Tokyo Chemical Industry Co., Ltd. (Tokyo, Japan). Butyl hydroxytoluene (BHT), nicotinamide adenine dinucleotide (NADH), and trichloroacetic acid (TCA) were purchased from Acros Organics (Geel, Belgium).

### 3.2. Preparation of M. fragrans Extract

The seeds of *M. fragrans* were collected from Wanhua Dist., Taipei City, Taiwan, in May 2020 and identified by Prof. J.-J. Chen. A voucher specimen was deposited in the Faculty of Pharmacy, National YangMing University, Taipei, Taiwan. Samples were collected, air-dried, and ground to powder. Two hundred microliters of various solvents (*n*-hexane, dichloromethane, chloroform, ethyl acetate, acetone, ethanol, and methanol) were added into 30 g of powder and incubated with shaking by orbital shakers for 24 h at 25 °C. The extracts were filtered through filter paper (Whatman No. 1) and concentrated under reduced pressure at 39 °C. All the extracts were stored at −20 °C until further use.

### 3.3. Preparation of Active Components

The seeds (30 g) of *M. fragrans* were extracted and pulverized three times with MeOH (200 mL each) for 3 days. The MeOH extract was condensed under reduced pressure at 35 °C, and the residue (fraction A, 5.46 g) was obtained. Fraction A (5.46 g) was purified by column chromatography (CC) (220 g of silica gel, 70–230 mesh; *n*-hexane/acetone gradient) to afford 10 fractions: A1–A10. Part (76 mg) of fraction A4 was further purified by preparative thin layer chromatography (TLC) (silica gel; dichloromethane/methanol, 19:1) to afford dehydrodiisoeugenol (10.3 mg) (R_f_ = 0.78). Part (112 mg) of fraction A5 was further by preparative TLC (silica gel; dichloromethane/methanol, 19:1) to obtain malabaricone B (6.9 mg) (R_f_ = 0.40) and malabaricone C (35.4 mg) (R_f_ = 0.31).

### 3.4. Normal-Phase HPLC

Normal-phase separations were performed using a Cosmosil 5SL-II column (5 μm; column of dimensions 10.0 × 250 mm) from Nacalai Tesque, Kyoto, Japan. High performance liquid chromatography-diode array detection (HPLC-DAD) chromatographic fingerprints were obtained with a LC-2000 Plus HPLC system (Jasco, Tokyo, Japan) equipped with a Jasco PU-2080 Plus pump, a G1379A micro vacuum degasser, a MD-2010 Plus diode array detector, and the ChromNAV software (version 2.0, Jasco, Tokyo, Japan) with LC-Net II/ADC system. Gradient separation using *n*-hexane (solvent A) and ethyl acetate (solvent B) as mobile phase was as follows: 0–10 min, linear gradient from 10 to 20% B; 10–30 min, 20% B with isocratic elution; 30–40 min, linear gradient from 20 to 30% B; 40–45 min, 30% B with isocratic elution; 45–48 min, linear gradient from 30 to 90% B; 48–50 min, 90% B with isocratic elution; 50–55 min, back to initial conditions at 10% B; and 55–60 min, at 10% B. The flow rate was 2.0 mL/min and the injection volume was 500 μL. Peaks were detected at 270 nm. Different compounds were identified by retention time. To guarantee peak purity, DAD acquisition from 200–650 nm was conducted to register UV-spectra. For the quantitative analysis of three compounds in the extracts, aliquots of samples were dispersed in 10 mL of an CH_2_Cl_2_/*n*-hexane (60/40, *v/v*) solution by sonication for 5 min. Then, the samples were centrifuged for 15 min at 3500 rpm, and the supernatant extracts were filtered through 0.45 μm polytetrafluoroethylene (PTFE) syringe filters (Zhejiang Sorfa Medical Plastic Co., Ningbo, China). Aliquots of filtrate were examined by the described HPLC conditions and calculated by standard curves. Quantification of three components from *M. fragrans* in each solvent extract followed as the method described previously [28]. The comparison of peak height with baseline noise level and a signal-to-noise ratio (*S/N*) of 3 and 10 under minimum concentration were defined as the limit of detection (LOD) and the limit of quantification (LOQ), respectively [29].

### 3.5. Reverse-Phase HPLC

Reversed-phase separations were performed using a Hypersil octadecyl silica (ODS) column (5 μm; column of dimensions 4.6 × 250 mm) purchased from Thermo Fisher Scientific, Inc. (Waltham, MA, USA). HPLC-photodiode array (PDA) detector chromatographic fingerprints were obtained with an Agilent 1260 Infinity II HPLC instrument equipped with a 1260 Infinity II quaternary pump, a 1260 Infinity II degasser, a 1260 Infinity II vialsampler, a 1260 Infinity II column thermostat, a 1260 Infinity II diode array detector HS, and a PC with the Agilent Chemstation software, all of them from Agilent Technologies (Waldbronn, Germany). Gradient separation using 0.1% formic acid in water (*v/v*) (solvent A) and isopropyl alcohol (solvent B) as mobile phase was as follows: 0–20 min, linear gradient from 30 to 40% B; 20–25 min, 40% B with isocratic elution; 25–45 min, linear gradient from 40 to 50% B; 45–50 min, 50% B with isocratic elution; 50–55 min, linear gradient from 50 to 98% B; 55-58 min, 98% B with isocratic elution; 58–60 min, back to initial conditions at 30% B; and 60–65 min, at 30% B. The flow rate was 0.8 mL/min and the injection volume was 15 μL. Peaks were detected at 270 nm. Different compounds were identified by retention time. To guarantee peak purity, DAD acquisition from 200–650 nm was conducted to register UV-spectra. For the quantitative analysis of three compounds in the extracts, aliquots of samples were dispersed in 10 mL of an isopropyl alcohol/water (60/40, *v/v*) solution by sonication for 5 min. Then, the samples were centrifuged for 15 min at 3500 rpm, and the supernatant extracts were filtered through 0.45 μm PTFE syringe filters (Zhejiang Sorfa Medical Plastic Co., Ningbo, China). Quantification of three components from *M. fragrans* in each solvent extract was as described above.

### 3.6. Determination of Total Phenolic Content

Total phenolic contents (TPCs) of different extracts were determined with slight modification of Folin-Ciocalteau’s method [30]. In brief, 50 μL of Folin-Ciocalteu reagent (0.5 N, dilute with deionized water) and 50 μL of each extract were mixed in a 96-well microplate and incubate for 5 min. Then, 100 μL of 20% sodium carbonate solution was added. The mixture was incubated for 45 min in the dark place at room temperature before measuring the absorbance of the supernatant at 750 nm after centrifugation for 8 min (1600× *g*). The TPC of the extracts were determined from a standard calibration curve using phloroglucinol. The concentration of TPC was expressed in mg gallic acid equivalents (GAE) per gram of dried extract. All measurements were conducted in triplicate.

### 3.7. DPPH Radical Scavenging Activity

The DPPH radical scavenging assay was measured as described with minor revisions [31]. The assay mixture in each well of the 96-well plates included 200 µM DPPH solution (dissolved in ethanol, 100 µL) and different concentrations of test compounds (100 µL). After 30 min at room temperature and in darkness, the absorbance of mixture was measured at 520 nm. DPPH radical scavenging activity was calculated with following formula,

DPPH scavenging activity (%) = (A_c_ − A_t_)/A_c_ × 100, where A_t_ is the absorbance of the test sample and A_c_ is the absorbance of the control. Commercially-available BHT was used as positive control. All half maximal inhibitory concentration (IC_50_) values of tested activities were determined by the liner regression of the percentage of remaining DPPH radical against the sample concentration.

### 3.8. ABTS Anion Radical Scavenging Activity

ABTS radical scavenging activity of each extract was measured as described earlier with slightly changes [32]. Briefly, ABTS solution was prepared by mixing 14 mM ABTS solution and 4.8 mM potassium persulfate (final concentration, 1/1, *v/v*) and leaving the mixture in the dark for about 16 h at room temperature. Ethanol was used to dilute the working solution for the absorbance to reach 0.700 ± 0.02 at 740 nm for measurements. Ten microliters of different concentrations of extract were added to 190 μL of ABTS solution. After reacting at room temperature for 6 min, the antioxidative activity of mixture was determined by calculating the decrease in absorbance measured at 732 nm by the following equation.

ABTS radical inhibiting activity (%) = (A_c_ − A_t_)/A_c_ × 100, where A_c_ and A_t_ are the absorbance of the control and test sample, respectively. The IC_50_ values of all tested activities were determined by the liner regression of the percentage of remaining ABTS radical against the sample concentration.

### 3.9. Superoxide Radical Scavenging Activity

Superoxide anion radical (O_2_^•−^) scavenging activity was measured using minor revision of method [33]. In brief, the stock solutions, which contained 300 μM NBT, 468 μM NADH, and 120 μM PMS, were added in 16 mM Tris-HCl buffer (pH 8.0). Each 50 μL of NBT, PMS, and different concentrations of extracts was mixed. The reaction was initiated by adding NADH solution to produce superoxide. The absorbance was measured at 560 nm after incubating at room temperature for 5 min. The scavenging activity was measured by the following equation.

Superoxide radical scavenging activity = (A_c_ − A_t_)/A_c_ × 100, where A_t_ is the absorbance of the test sample and A_c_ is the absorbance of the control. The IC_50_ values of all tested activities were determined by the liner regression of the percentage of remaining superoxide radical against the sample concentration.

### 3.10. Hydroxyl Radical Scavenging Activity

The scavenging activity of hydroxyl radical was evaluated based on the method described by Mathew and Abraham with slight modification [34]. Briefly, the reaction mixture, contained different concentrations of the extract, 0.1 mM FeCl_3_, 0.1 mM EDTA, 2.8 mM deoxyribose, 0.1 mM ascorbic acid, and 1 mM H_2_O_2_ in KH_2_PO_4_–KOH buffer (20 mM pH 7.4), was incubated in a water bath at 37 °C for the nonsite-specific hydroxyl radical system. After incubation of 1 h, 2.8% TCA (200 μL) and 1% TBA (200 μL) were added. After boiling at 100 °C for 30 min, the antioxidative activity of the boiled mixture was measured by calculating the decrease in absorbance measured at 532 nm by the following equation.

Hydroxyl radical scavenging activity (%) = (A_c_ − A_t_)/A_c_ × 100, where A_t_ and A_c_ are the absorbance of the test sample and the control, respectively. The IC_50_ values of all tested activities were determined by the linear regression of the percentage of remaining hydroxyl radical against the sample concentration.

### 3.11. α-Glucosidase Inhibitory Activity Assay

The inhibition assay of α-glucosidase was performed using the conditions previously reported with slight modifications [35]. In brief, the solution of α-glucosidase (1 U/mL in 0.1 M phosphate buffer, pH 6.8; 20 μL) was mixed with different concentration of tested compounds (20 μL) in a 96-well microplate. Then, the substrate *p*-nitrophenyl-α-d-glucopyranoside (*p*-NPG) (0.375 mM; 40 μL) was added and incubated at 37 °C for 30 min under shaking in the microplate. After adding 0.1 M Na_2_CO_3_ solution (80 μL), the reaction was terminated, and the absorbance of *p*-NPG was measured at 405 nm using a 96-well microplate reader. Each experiment was executed in triplicate. Percentage inhibition of each sample was determined using the following equation,

α-Glucosidase inhibition (%) = (A_c_ − A_t_)/A_c_ × 100, where A_c_ is the absorbance of the control, and A_t_ is the absorbance of the test sample. The IC_50_ values of all tested activities were determined by the liner regression of the percentage of remaining α-glucosidase against the sample concentration.

### 3.12. Statistical Analysis

All data are expressed as mean ± SEM. Statistical analysis was carried out using Student’s *t*-test. A probability of 0.05 or less was considered statistically significant. All the experiments were performed at least 3 times.

## 4. Conclusions

Various solvent extracts of *M. fragrans* were investigated with various antioxidant systems and anti-α-glucosidase activity assay. TPC in the extracts of methanol and ethanol showed approximately five to six times the amount of TPC in those using *n*-hexane, dichoroform, and chloroform, proving that suitable relative polarity of extracting solvents for TPC from *M. fragrans* would be ranged from 0.654 to 0.762. In our study, methanol, ethanol, and acetone extracts exhibited strong DPPH, ABTS, and hydroxyl scavenging activities, which may be aligned with TPC in the extracts. Methanol extract also possessed the highest antioxidant activity among all the other solvent extracts. Furthermore, methanol extracts also showed strong anti-α-glucosidase activity. The bioactivity assays demonstrated that dehydrodiisoeugenol, malabaricone B, and malabaricone C displayed antioxidant activities, and malabaricone C showed strong anti-α-glucosidase effect. 

In conclusion, this study showed that extraction solvents for *M. fragrans* affected extraction yield, antioxidant activities, and bioactive component levels. The methanol, ethanol, and acetone extracts displayed relatively high TPC level and antioxidant activities. The methanol extract contained the largest amount of polyphenols among extracts. The active antioxidant components in *M. fragrans* were identified as dehydrodiisoeugenol, malabaricone B, and malabaricone C based on their antioxidant activities. The above active extracts and their isolates can be used as natural antioxidants in the food industry, as well as dietary supplement against oxidative damage. Furthermore, the methanol and ethanol extracts can also be used as natural α-glucosidase inhibitors.

## Figures and Tables

**Figure 1 molecules-25-05198-f001:**
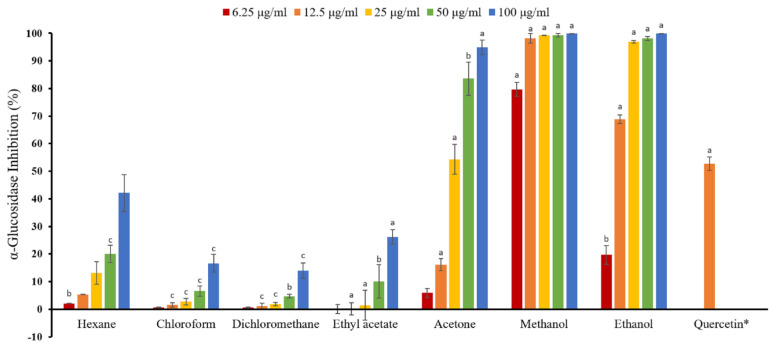
Bars show α-glucosidase inhibition at different concentrations of solvent extracts from *M. fragran.* * Quercetin (12.5 μg/mL) was used as a positive control. Bars represent mean ± SEM of three experiments. Bars marked with different letters are significantly different (^a^
*p* < 0.001; ^b^
*p* < 0.01; ^c^
*p* < 0.05 compared to control).

**Figure 2 molecules-25-05198-f002:**
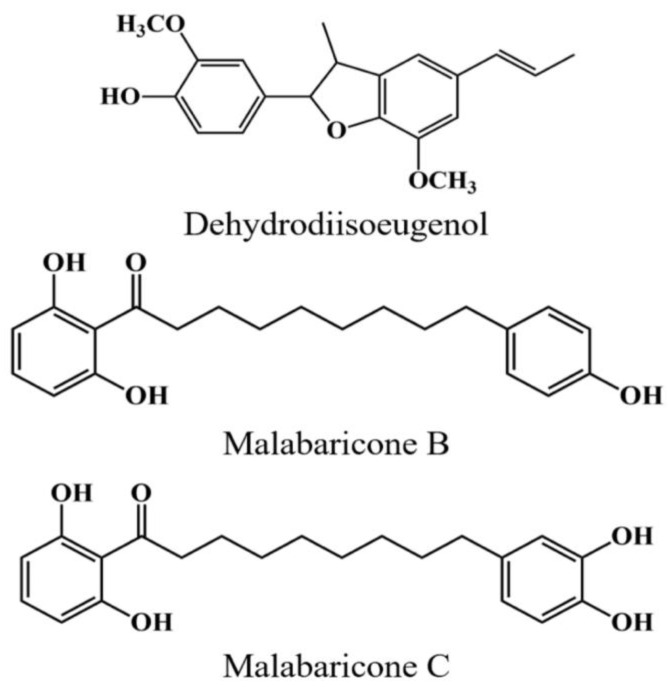
Chemical structures of three compounds from *M. fragrans*.

**Figure 3 molecules-25-05198-f003:**
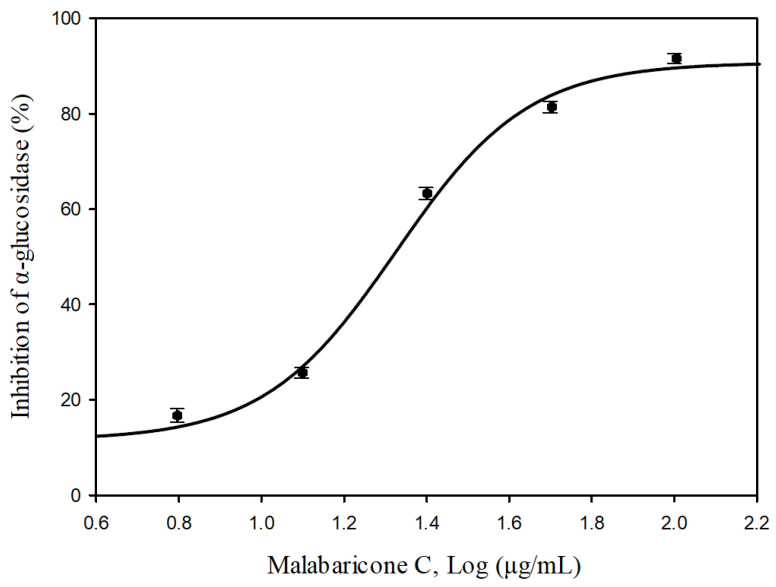
Dose-response (IC_50_) curve of malabaricone C. Each point represents the average of triplicate measurements.

**Table 1 molecules-25-05198-t001:** Total phenol contents and extraction yields of *Myristica fragrans* with each extraction solvent.

Extracting Solvents	Relative Polarity	TPC (mg/g) ^a^ (GAE)	Yields (%) ^b^
*n*-Hexane	0.009	16.82 ± 0.62 ***	27.3 ± 1.67
Chloroform	0.259	18.65 ± 0.53 ***	29.2 ± 0.79
Dichloromethane	0.269	18.97 ± 1.22 **	30.7 ± 1.49
Ethyl acetate	0.288	32.93 ± 0.85 ***	24.5 ± 1.13
Acetone	0.355	70.07 ± 2.28 ***	21.1 ± 0.23
Methanol	0.762	107.83 ± 0.66 ***	18.2 ± 0.75
Ethanol	0.654	98.01 ± 2.99 ***	15.6 ± 1.21

^a^ Total Phenolic Content (TPC) was expressed in mg of gallic acid equivalents (GAE) per gram of extract. Values are expressed as means ± standard error; ^b^ Yield was calculated as % yield = (weight of extract/initial weight of dry sample) × 100; ** *p* < 0.01 compared with the control; *** *p* < 0.001 compared with the control.

**Table 2 molecules-25-05198-t002:** The antioxidant activities of different solvent extracts from *Myristica fragrans* determined with 2,2-diphenyl-1-(2,4,6-trinitrophenyl)hydrazyl (DPPH), 2,2′-azino-bis(3-ethylbenzothiazoline-6-sulfonic acid) (ABTS), superoxide, and hydroxyl radicals.

Extracting Solvents	DPPH IC_50_ (μg/mL)	ABTS IC_50_ (μg/mL)	Superoxide IC_50_ (μg/mL)	Hydroxyl IC_50_ (μg/mL)
*n*-Hexane	126.57 ± 6.23 *	103.05 ± 2.41 *	>400	51.94 ± 0.79 *
Chloroform	167.17 ± 7.13	93.70 ± 5.06 *	>400	82.39 ± 2.62 *
Dichloromethane	96.90 ± 7.68	82.31 ± 2.15 *	>400	88.19 ± 2.09 *
Ethyl acetate	95.12 ± 2.63 *	91.19 ± 0.88 *	>400	55.25 ± 1.25 *
Acetone	65.08 ± 1.44 *	64.35 ± 1.58 *	>400	42.99 ± 0.19 *
Methanol	22.42 ± 0.99 **	34.41 ± 0.78 **	117.66 ± 2.56 *	37.81 ± 1.56 *
Ethanol	39.65 ± 0.83 *	27.68 ± 0.31 **	>400	56.05 ± 2.52 *
BHT ^a^	36.94 ± 0.49 **	11.05 ± 0.26 **	N.A. ^b^	61.51 ± 2.46 *

Results are expressed as half inhibitory concentration (IC_50_) of each free-radical scavenging activity; ^a^ Butylated hydroxytoluene (BHT) used as positive control; ^b^ N.A. indicates not available; * *p* < 0.05 and ** *p* < 0.01 compared with the control.

**Table 3 molecules-25-05198-t003:** α-Glucosidase inhibitory activities of different solvent extracts.

Extracting Solvents	α-Glucosidase IC_50_ (μg/mL)
*n*-Hexane	>200
Chloroform	>200
Dichloromethane	>200
Ethyl acetate	185.36 ± 5.21
Acetone	29.07 ± 2.30 *
Methanol	4.08 ± 0.12 **
Ethanol	11.92 ± 0.39 *
Quercetin ^a^	14.99 ± 0.81 **

^a^ Quercetin used as positive control; * *p* < 0.05 and ** *p* < 0.01 compared with the control.

**Table 4 molecules-25-05198-t004:** Identification and quantification of the major active components from *Myristica fragrans* in different solvent extracts.

Extracting Solvents	Malabaricone B (mg/g)	Malabaricone C (mg/g)	Dehydrodiisoeugenol (mg/g)	Total Amount (mg/g)
Methanol	6.17 ± 0.51	31.67 ± 1.49	13.59 ± 0.50	51.43 ± 1.18
Ethanol	4.65 ± 0.54	27.54 ± 1.16	10.61 ± 0.59	42.80 ± 1.17
Acetone	2.72 ± 0.13	16.41 ± 0.91	6.62 ± 0.19	25.75 ± 0.67
Ethyl acetate	2.29 ± 0.28	15.12 ± 0.67	5.86 ± 0.89	23.27 ± 1.72
Chloroform	2.50 ± 0.05	4.48 ± 0.27	11.27 ± 0.54	18.25 ± 0.65
Dichloromethane	2.58 ± 0.08	3.89 ± 0.59	10.18 ± 0.42	16.65 ± 0.92
*n*-Hexane	1.10 ± 0.13	N.D. ^a^	14.40 ± 0.36	15.52 ± 0.26

Results are expressed as milligrams of each compound in gram of extract; ^a^ N.D. indicates not detected.

**Table 5 molecules-25-05198-t005:** The antioxidant activities of isolated components from *Myristica fragrans* determined with DPPH, ABTS, superoxide, and hydroxyl.

Compounds	DPPH IC_50_ (μg/mL)	ABTS IC_50_ (μg/mL)	Superoxide IC_50_ (μg/mL)	Hydroxyl IC_50_ (μg/mL)
Dehydrodiisoeu-genol	66.02 ± 2.85 *	8.43 ± 0.42 ***	>200	68.29 ± 0.70
Malabaricone B	>200	7.05 ± 0.72 ***	>200	95.22 ± 4.20
Malabaricone C	8.35 ± 2.20 **	5.36 ± 0.19 **	>200	72.81 ± 2.58 *
BHT ^a^	34.28 ± 1.40 *	10.67 ± 0.41 **	N.A. ^b^	69.96 ± 4.66 *

Results are expressed as half inhibitory concentration (IC_50_) of each free-radical scavenging activity. ^a^ Butylated hydroxytoluene (BHT) used as positive control; ^b^ N.A. indicates not available; * *p* < 0.05, ** *p* < 0.01, and *** *p* < 0.001 compared with the control.

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
