# Peer review of "Evaluation of Antioxidant and Anti-α-glucosidase Activities of Various Solvent Extracts and Major Bioactive Components from the Seeds of Myristica fragrans"

_molecules, 2020, doi:10.3390/molecules25215198_

Round 1
Reviewer 1 Report
The manuscript is well written and is concluded precisely. A large amount of work was involved in the study, and as far as I can determine, the work is solid. Below are some specific points.
- The selected concentration range of solvent extracts from M. fragrans should be explained. Was a cytotoxicity test performed (e.g SRB)?
- Figure 1 bars were represented as mean ± standard deviation of three experiments. The authors have stated that all of the data were expressed as mean ± SEM in the statistical analysis section. Please check it ?
Author Response
Detailed Responses for the Reviewer #1’s comments
Ms. Ref. No.: molecules-989113 November 03, 2020
Title: Evaluation of antioxidant and anti-α-glucosidase activities of various solvent extracts and major bioactive components from Myristica fragrans
Comment and suggestions for authors:
The manuscript is well written and is concluded precisely. A large amount of work was involved in the study, and as far as I can determine, the work is solid. Below are some specific points.
Responses:
Thank you very much for carefully reviewing our manuscript and kindly offering your suggestions. We have explained your remarks as the following statements:
Comment 1:
The selected concentration range of solvent extracts from M. fragrans should be explained. Was a cytotoxicity test performed (e.g SRB)?
Responses:
In our selected concentration range, most of solvent extracts have shown significant bioactivities and the results are concentration-dependent. These results indicated that the range we've selected is suitable for the experiment.
Since our experiments didn’t use cells, the cytotoxicity test was not performed. Besides, the previous research has shown that ethanolic extract of M. fragrans seed was non-toxic to RAW264.7 cells at the concentration of 100 μg/mL [K. Dewi, et al., 2015]. Furthermore, M. fragrans has become both food and medicine in some countries, especially in India and China. [J. Asgarpanah and N. Kazemivash, 2012]. So we suggested that M. fragrans would not cause serious side effects under appropriate concentration.
Reference:
Dewi, K.; Widyarto, B.; Erawijantari, P.P.; Widowati, W. In vitro study of Myristica fragrans seed (Nutmeg) ethanolic extract and quercetin compound as anti-inflammatory agent. International Journal of Research in Medical Sciences 2015, 3, 2303–2310.
Asgarpanah, J.; Kazemivash, N. Phytochemistry and pharmacologic properties of Myristica fragrans Hoyutt.: A review. African Journal of Biotechnology 2012, 11, 12787–12793.
Comment 2:
Figure 1 bars were represented as mean ± standard deviation of three experiments. The authors have stated that all of the data were expressed as mean ± SEM in the statistical analysis section. Please check it ?
Responses:
Figure 1 bars have been corrected and represented as mean ± SEM of three experiments, in accordance with the reviewer’s comments.

Reviewer 2 Report
The authors reports studies on the evaluation of antioxidant and alpha-glucosidase activity of extracts from Myristica fragrans. The studies are well performed and the results seem convincing. Minor revision to expand the discussion of the results would enhance the report.
- In Table 1, the authors show that methanol had the highest yield of TPC, which was 5-6 times greater than dichloromethane. On the other hand, dichloromethane had apparently the highest yield overall. This difference should be discussed and explained.
- The authors show in Table 2 that methanol or ethanol extracts have the highest DPPH and ABTS scavenging activity, but little superoxide scavenging activity. The reasons for this difference should be discussed. What is the proposed mechanism for free-radical scavenging activity that could explain these differences? Similarly, the results shown in Table 5 require a similar discussion.
- The author propose extracts of M. fragrans could be used to control blood sugar without side effects. Further discussion should be provided regarding this suggestion. Is there evidence that extracts of M. fragrans lack GI side effects that are seen with most other alpha-glucosidase inhibitors? Is there any reported data on the glucose lowering effects of M. fragrans extracts? Additional discussion of the importance or significance of the reported studies would greatly enhance this manuscript.
- Minor comment: There are a number of typographical errors that need to be corrected.
Author Response
Please see an attached file.

Reviewer 3 Report
In this work authors present antioxidant properties (using DPPH, ABTS, superoxide radical, and hydroxyl radical scavenging tests) and anti-α-glucosidase activities of five different solvent extracts from Myristica fragrans and its major bioactive components.
Since this paper could be of interest for the scientific community, I'd recommend publishing it in " Molecules" after some minor revisions of the manuscript:
- on pg.7, in the Chapter 2.10. “Anti-α-glucosidase activities of isolated component“: Why did the authors choose only malabaricone C for the analysis of inhibitory activity against α-glucosidase? Does this mean that the other two isolated components (malabaricone B and dehydrodiisoeugenol) exhibited weaker inhibition of α-glucosidase compared to malabaricone C? The manuscript should state the reason why only malabaricone C was chosen!
- on 9, line 264: Abbreviation “TPC” should be defined in parentheses the first time they appear in the text (pg.2, line 75)!
- on pg.9, in the Chapter 6. „Determination of total phenolic content“ :Why did the authors express the concentration of TPC in gallic acid equivalents, if they used phloroglucinol for the standard calibration curve?
- I suggest to the authors to indicate: a) in the title of the manuscript, b) in the abstract, and c) in the last sentence of the Introduction section that it is a Myristica fragrans seeds.
Author Response
Comment and suggestions for authors:
In this work authors present antioxidant properties (using DPPH, ABTS, superoxide radical, and hydroxyl radical scavenging tests) and anti-α-glucosidase activities of five different solvent extracts from Myristica fragrans and its major bioactive components.
Since this paper could be of interest for the scientific community, I'd recommend publishing it in "Molecules" after some minor revisions of the manuscript:
Responses:
Thank you very much for carefully reviewing our manuscript and kindly offering your suggestions. We have explained your remarks as the following statements:
Comment 1:
On pg.7, in the Chapter 2.10. “Anti-α-glucosidase activities of isolated component“: Why did the authors choose only malabaricone C for the analysis of inhibitory activity against α-glucosidase? Does this mean that the other two isolated components (malabaricone B and dehydrodiisoeugenol) exhibited weaker inhibition of α-glucosidase compared to malabaricone C? The manuscript should state the reason why only malabaricone C was chosen!
Responses:
We only chose malabaricone C for the analysis of inhibitory activity against α-glucosidase because the inhibitory activities of other compounds were weak and not significant. Therefore, We only chose malabaricone C for the analysis of inhibitory activity against α-glucosidase.
We had added at the section 2.10. “With further analysis of the α-glucosidase inhibitory activity, we conduct further investigation on each components isolated from M. fragrans. Among the isolated compounds, we found that malabaricone C possesses strong inhibitory activity against α-glucosidase.”
Comment 2:
On 9, line 264: Abbreviation “TPC” should be defined in parentheses the first time they appear in the text (pg.2, line 75)!
Responses:
The abbreviation of TPC has been defined, in accordance with the reviewer’s comments.
Comment 3:
On pg.9, in the Chapter 6. “Determination of total phenolic content” :Why did the authors express the concentration of TPC in gallic acid equivalents, if they used phloroglucinol for the standard calibration curve?
Responses:
According to the previous study [Ford, L., et al., 2019], phloroglucinol is the most widely used equivalent method to use when comparing between seaweed species, however, the majority of studies analyzing terrestrial plants use gallic acid equivalents. When using phloroglucinol for the standard calibration curve for M. fragrans, a calibration of both phloroglucinol and gallic acid need to be conducted in order to compare the phenolic content between different terrsertrial species.
Reference: Ford, L., Theodoridou, K., Sheldrake, G. N., Walsh, P. J. A critical review of analytical methods used for the chemical characterisation and quantification of phlorotannin compounds in brown seaweeds. Phytochemical Analysis, 2019, 30, 587-599.
Comment 4:
I suggest to the authors to indicate: a) in the title of the manuscript, b) in the abstract, and c) in the last sentence of the Introduction section that it is a Myristica fragrans seeds.
Responses:
These have been corrected, in accordance with the reviewer’s comments.

Reviewer 4 Report
This manuscript describes the extraction and the characterization of three natural compounds obtained from Myristica fragrans. In particular, methanol extracts of these molecules are found as good source of natural antioxidant and α-glucosidase inhibitor. The contribution is simple but well-structured and provides interesting data on a relatively unexplored aspects of molecules contained in this spice.
I have just some minor remarks:
Line 18-19: Please define the abbreviations DPPH and ABTS. You can do this here (abstract) or in the result section
Line 51: “importance” should be “important”
Line 76: Please define the abbreviation TPC
Line 160: “compare” should be “compared”
Line 208 and 235: Please use the abbreviation “g” for grams and “min” for minutes
Author Response
Detailed Responses for the Reviewer #4’s comments
Ms. Ref. No.: molecules-989113 November 02, 2020
Title: Evaluation of antioxidant and anti-α-glucosidase activities of various solvent extracts and major bioactive components from Myristica fragrans
Comment and suggestions for authors:
This manuscript describes the extraction and the characterization of three natural compounds obtained from Myristica fragrans. In particular, methanol extracts of these molecules are found as good source of natural antioxidant and α-glucosidase inhibitor. The contribution is simple but well-structured and provides interesting data on a relatively unexplored aspects of molecules contained in this spice.
I have just some minor remarks:
Responses:
Thank you very much for carefully reviewing our manuscript and kindly offering your suggestions. We have explained your remarks as the following statements:
Comment 1:
Line 18-19: Please define the abbreviations DPPH and ABTS. You can do this here (abstract) or in the result section.
Responses:
The abbreviations, DPPH and ABTS, have been defined, in accordance with the reviewer’s comments.
Comment 2:
Line 51: “importance” should be “important”
Responses:
This has been corrected, in accordance with the reviewer’s comments.
Comment 3:
Line 76: Please define the abbreviation TPC
Responses:
TCP has been defined as ‘total phenolic content’, in accordance with the reviewer’s comments.
Comment 4:
Line 160: “compare” should be “compared”
Responses:
This has been corrected, in accordance with the reviewer’s comments.
Comment 5:
Line 208 and 235: Please use the abbreviation “g” for grams and “min” for minutes
Responses:
These have been corrected, in accordance with the reviewer’s comments.

Reviewer 5 Report
This research manuscript reported the influence of extraction solvents on M. fragrans on their antioxidant activities. Also, the author isolated the bioactive compounds from M. fragrans and revealed the higher antioxidant capacity of malabaricone B compound. The manuscript is presented well with the required analysis. I have the following queries and suggestions for the authors:
- The authors should mention the abbreviations in the introduction. For instance, TPC (Total Phenolic Content) was mentioned on page 9, line 264, IC50 (half inhibitory concentration) was mentioned on page 4, line 121.
- The authors described only the results in the ‘Results and Discussion’ section and the related discussion similar research articles is missing.
- Most articles cited in this manuscript were published in the year 2000-2010. Also many cited articles are not related to M. frangrans.
- Detailed isolation of bioactive of using GC-MS from M. frangrans:
- https://doi.org/10.1016/j.indcrop.2018.12.064;
- https://www.mdpi.com/1420-3049/24/6/1062/htm
Author Response
Detailed Responses for the Reviewer #5’s comments
Ms. Ref. No.: molecules-989113 November 03, 2020
Title: Evaluation of antioxidant and anti-α-glucosidase activities of various solvent extracts and major bioactive components from Myristica fragrans
Comment and suggestions for authors:
This research manuscript reported the influence of extraction solvents on M. fragrans on their antioxidant activities. Also, the author isolated the bioactive compounds from M. fragrans and revealed the higher antioxidant capacity of malabaricone B compound. The manuscript is presented well with the required analysis. I have the following queries and suggestions for the authors:
Responses:
Thank you very much for carefully reviewing our manuscript and kindly offering your suggestions. We have explained your remarks as the following statements:
Comment 1:
The authors should mention the abbreviations in the introduction. For instance, TPC (Total Phenolic Content) was mentioned on page 9, line 264, IC50 (half inhibitory concentration) was mentioned on page 4, line 121.
Responses:
These have been corrected, in accordance with the reviewer’s comments.
The abbreviation is mentioned when it first appears in the text.
Comment 2:
The authors described only the results in the ‘Results and Discussion’ section and the related discussion similar research articles is missing.
Responses:
More related discussions have been added in section 2.1, 2.5, 2.6, 2.7, 2.9, and 2.10, in accordance with the reviewer’s comments.
Comment 3:
Most articles cited in this manuscript were published in the year 2000-2010. Also many cited articles are not related to M. frangrans.
Detailed isolation of bioactive of using GC-MS from M. frangrans: https://doi.org/10.1016/j.indcrop.2018.12.064;
https://www.mdpi.com/1420-3049/24/6/1062/htm
Responses:
Some articles cited in this manuscript have been corrected, in accordance with the reviewer’s comments.
We have corrected the first sentence at the section 2.7 as “In previous studies, phenolic derivatives from M. fragrans were well identified and separated by HPLC with normal- and reverse-phase column chromatograph [23,24].”
[23] Saputri, F.A.; Lestari, K.; Levita, J. Determination of safrole in ethanol extract of Nutmeg (Myristica fragrans Houtt) using reversed-phase high performance liquid chromatography. Int. J. Chem., 2014, 6, 14-20.
[24] Chiu, S.; Wang, T.; Belski, M.; Abourashed, E. A. HPLC-guided isolation, purification and characterization of phenylpropanoid and phenolic constituents of nutmeg kernel (Myristica fragrans). Nat. Prod. Commun., 2016, 11, 483-488.

Round 2
Reviewer 5 Report
Thank you for providing the revised manuscript.